# Occurrence of Leishmaniasis in Iberian Wolves in Northwestern Spain

**DOI:** 10.3390/microorganisms11051179

**Published:** 2023-04-30

**Authors:** Javier Merino Goyenechea, Verónica Castilla Gómez de Agüero, Jesús Palacios Alberti, Rafael Balaña Fouce, María Martínez Valladares

**Affiliations:** 1Departamento de Ciencias Biomédicas, Facultad de Veterinaria, Universidad de León, Campus de Vegazana s/n, 24071 León, Spain; 2Centro del Lobo Ibérico “Félix Rodríguez de la Fuente” Robledo De Sanabria, 49393 Puebla De Sanabria, Spain; jpalaciosalberti@gmail.com; 3Departamento de Sanidad Animal, Facultad de Veterinaria, Universidad de León, Campus de Vegazana s/n, 24071 León, Spain; vcastg@unileon.es (V.C.G.d.A.); mmarva@unileon.es (M.M.V.); 4Instituto de Ganadería de Montaña, CSIC—Universidad de León, Finca Marzanas. Ctra. León-Grulleros s/n, 24346 León, Spain

**Keywords:** wild fauna, Iberian wolf, canine leishmaniasis, diagnosis, PCR, buccal samples, hair, northwestern Spain

## Abstract

Canine leishmaniasis is an important vector-borne protozoan disease in dogs that is responsible for serious deterioration in their health. In the Iberian Peninsula, as in most countries surrounding the Mediterranean Sea, canine leishmaniasis is caused by *Leishmania infantum* (zymodeme MON-1), a digenetic trypanosomatid that harbors in the parasitophorous vacuoles of host macrophages, causing severe lesions that can lead to death if the animals do not receive adequate treatment. Canine leishmaniasis is highly prevalent in Spain, especially in the Mediterranean coastal regions (Levante, Andalusia and the Balearic Islands), where the population of domestic dogs is very high. However, the presence of this disease has been spreading to other rural and sparsely populated latitudes, and cases of leishmaniasis have been reported for years in wildlife in northwestern Spain. This work describes for the first time the presence of wolves that tested positive for leishmaniasis in the vicinity of the Sierra de la Culebra (Zamora province, northwestern Spain), a protected sanctuary of this canid species, using PCR amplification of *L. infantum* DNA from different non-invasive samples such as buccal mucosa and those from both ears and hair. In addition to live animals (21), samples from carcasses of mainly roadkill animals (18) were also included and analyzed using the same technique, obtaining a positivity rate of 18 of the 39 wolves sampled (46.1%) regardless of their origin.

## 1. Introduction

Canine leishmaniasis (CanL) is a re-emerging vector-borne disease produced by the protist parasite *Leishmania infantum* [1,2]. CanL is endemic in the coastal regions surrounding the Mediterranean basin, and it is increasingly spreading to the northern areas of southern European countries, including Spain, with a cold continental climate in winter and mild temperatures in spring and summer [3]. Currently, the vectors of the disease (dipteran species of the genus *Phlebotomus* or sand flies) have colonized almost the entire Iberian Peninsula, and therefore, CanL is being diagnosed more frequently in areas previously described as non-endemic [4]. CanL is usually lethal in untreated dogs, and lesions are associated with inflammatory reactions including peripheral lymphadenopathy and hepatosplenomegaly. Dermatological signs include alopecia, dermatitis, onychogryphosis and epistaxis. Renal disease is severe and may progress from nephrotic syndrome to chronic renal failure [2,5,6].

CanL is endemic in all European countries bordering the Mediterranean Sea, but the distribution of the disease is heterogeneous between and within countries [7]. In Greece, the mean seropositivity to CanL in dogs was 22.1%. In the Balkan countries, seropositivity ranged from 1.8–10.6% [8]. In Italy, a canine population prevalence of 10–40% was detected in the Campania region [9], whereas in France, the highest prevalence (between 8.1 and 28%) was found in areas closer to the Mediterranean Sea [10]. In Spain, unlike Portugal, the lowest figures have been found in inland and northern regions, whereas in Levante, Andalusia and the Balearic Islands, the average seropositivity is higher than 17% [11,12]. The spread of the disease to pets in non-endemic northern European countries [13] has been attributed to several factors including global warming, pet tourism and pet trade [14,15].

The first description of this infection in wildlife was made by Rioux and coworkers, who found several cases of leishmaniasis in red foxes (*Vulpes vulpes*) in southern France [16]. The incorporation of PCR as a diagnostic technique has facilitated the detection of wildlife animals that have been in contact with the parasite in some way, confirmed in many cases by the use of complementary immunological and parasitological techniques. In addition to foxes, many other mammals have tested positive for leishmaniasis using this technique, including wild canids [17,18], felids such as lynxes, wild cats, mustelids [19,20,21], hares, rabbits, bats and rodents [22,23,24], among others.

Wolves are expected to be potential reservoirs of the pathogen due to their phylogenetic similarity to dogs, their social habits and their proximity to rural areas, where dogs continue to play important domestic, guarding, shepherding and hunting roles [25]. Since the first description of leishmaniasis in wolves in Croatia in 2003—reported by Beck and coworkers in 2008 [26]—studies on the existence of this disease have been spurious, unsystematic and, in most cases, carried out on the remains of accidentally killed or poisoned animals [27,28,29].

Finally, the importance of CanL should not only be addressed from the point of view of the suffering of infected animals, but also as a real public health problem that may worsen with global warming. Therefore, it is of great importance to prevent infection as much as possible and eventually eradicate the disease, which implies close collaboration between authorities and health professionals. In the present work, we describe the presence, by means of semiquantitative PCR, of *L. infantum* DNA in buccal mucosa, ears and hair taken from both animal carcasses (found by environmental protection services) and live Iberian wolves (*Canis lupus signatus*) housed in a semicaptive wolf reserve located in the province of Zamora (Castilla y León, northwestern Spain).

## 2. Materials and Methods

### 2.1. Origin of the Animals

The study was carried out in the province of Zamora (Castilla y León, Spain) in northwestern Spain (see map in Figure 1) with two well-characterized climate types: a humid climate in the mountainous area (the northwestern area of the province), with abundant rainfall, cold winters, snowfall and mild summers; an extreme continental climate, which affects the rest of the province, with cold winters and hot summers. At present, the wolf occupies practically the entire territory of Zamora (Figure 1); however, there is a mountain range called “Sierra de La Culebra” that has the highest density of Iberian wolves in the Iberian Peninsula. This is where they cohabit with other species of carnivores [30]. In the last Regional Census elaborated by Sáenz de Buruaga and coworkers between 2012 and 2013, 179 wolf packs were identified in the Community of Castilla y León, of which 152 were located to the north of the Duero River and 27 to the south. The province of Zamora was assigned 45 of these packs [31,32].

A key enclave for protection of the Iberian wolf, named “Centro del Lobo Ibérico Félix Rodríguez de la Fuente” (Iberian Wolf Center) [33], is located in northwest Zamora. It maintains a stable population of 14 wolves kept in semi-freedom and distributed among three established packs. The wolves are kept in an ecosystem similar to that of the Sierra de La Culebra. The wolves at the Iberian Wolf Center are housed in three different enclosures, interacting with each other and also with members of the management team.

### 2.2. Sampling

Samples were collected from 39 Iberian wolves (*Canis lupus signatus*) in the province of Zamora. Some of the samples came from live animals captured and tagged as part of the Iberian wolf management plan in Castilla y León (7), and others (14) were obtained at the Iberian Wolf Center’s “Félix Rodríguez de la Fuente”. Other samples were collected from wolves killed by natural causes (1), motor vehicles (14) or from legal hunting activity (wolf hunting in Spain was authorized north of the Duero River until 2021) carried out between the years 2017 and 2021 (3).

Animals either captured in the wildness or those living in semicaptivity in the Iberian Wolf Center were subjected to an exhaustive protocol established by the Management and Veterinary Care Unit before sampling. After capture, the first action of the veterinary team consisted of sedation with a mixture of ketamine (15 mg/kg body weight) and medetomidine hydrochloride (30 μg/kg body weight) via intramuscular injection using a blowgun. During the 20 min of sedation, a GPS geolocation collar was placed, and biometric measurements were taken.

Samples of buccal mucosa, ears and hair were taken from wolves. The first two samples were taken with the help of a sterile swab; for the buccal mucosa, a scraping of the inside of the cheek was performed to collect as many buccal cells as possible. Ear samples were collected by scraping the epithelial mucosa of the inner ear. Regarding hair collection, individual hairs were taken, including the hair follicle, from the heads of the animals. After sample collection, all samples were sent refrigerated to the laboratory for analysis. Once in the laboratory, they were stored at −20 °C until DNA extraction.

Sampling of the wolf carcasses was done after the necropsy and also included buccal mucosa, ear swabs and hairs, collected as previously described. Additionally, samples of spleen, liver, kidney, muscle, skin and blood from cavities and urine were also routinely obtained. 

### 2.3. PCR Analyses of the Samples

DNA extraction from all samples was performed following the protocols described in the GeneJet genomic DNA purification kit (Thermo Scientific™, Waltham, MA, USA) for samples collected with a swab or directly from tissue following the manufacturer’s instructions. The final DNA elution volume was 25 μL per sample. As a positive control, DNA extraction from a culture of *L. infantum* promastigotes was also performed.

To analyze the parasite load in the animal’s hair, initially, groups of 5, 10 and 20 individual hairs were taken from a positive wolf, and DNA extraction was performed on these three groups independently. For the rest of the hair samples, the extraction was done using 10 hairs from the head of the same wolf.

All samples were analyzed via qualitative PCR after amplification of a 131 base pair (bp) fragment of the kinetoplast minicircle of *L. infantum* using the newly synthesized primers (Forward: 5′-CCCAAACTTTTCTGGTCCTC-3′; Reverse: 5’-TTACACCAACCCCCAGTTTC-3′) corresponding to a conserved region of DNA minicircle of *L. infantum* kinetoplast. As an internal control, to confirm that cells had been collected with the swab, a fragment of 303 bp of the gene coding the Na^+^/Ca^2+^ exchanger (NCX1) was also amplified in the buccal mucosa and inner ear samples using the pair of primers previously described [34] (Forward 5′-CCTAGGTCTCCTGCAGTGAAGT-3′; Reverse: 5′-CCAAGACCCTTCCTTTGGA-3′). The PCR amplification was carried out using DNA AmpliTools HotSplit Master Mix (BioTools, Jupiter, FL, USA) in a final volume of 20 µL, and the primer concentration was 0.5 µM each. Regarding the DNA volume added for each reaction, 5 µL and 1 µL of DNA were added for the amplification of *L. infantum* and the NCX1 gene, respectively. The thermocycler used (Biorad, Hercules, CA, USA) was set to run for 5 min at 95 °C, followed by 40 cycles each of 30 s at 95 °C, 30 s at 60 °C, 30 s at 72 °C and finally, an extension step at 72 °C for 5 min. Amplification products were analyzed via electrophoresis in 2% agarose TBE (Tris base, boric acid and EDTA). The specific amplified fragments were purified using a SpeedTools Clean-up kit (BioTools), and Sanger sequencing was done at the University of León Sequencing Facility to confirm the specificity of the PCR.

## 3. Results

### 3.1. Set-Up of a PCR-Based Method for L. infantum DNA in Wolf Hair

Wolf hair is a non-traumatic, easy and safe place to collect samples from wild animals. L. infantum amplification in wolf hair samples was determined by extracting DNA for PCR amplification using increasing numbers of hairs (5, 10 and 20). 

As shown in Figure 2, a quantity-dependent 131 bp band was amplified from *L. infantum* DNA when the pair of primers corresponding to the kinetoplast DNA were used. This figure shows that the amplified band was observed from 10 hairs onwards; therefore, with the rest of the samples, it was decided to analyze a total of 10 hairs per animal. In the same Figure, a 303 bp band encoding *Canis lupus* NCX1 gene was also amplified, showing its validity to be used for the collection of wild animal samples.

### 3.2. Occurrence of Leishmaniasis in Living Wolves

The present study was conceived to address the presence of *L. infantum* in live captured or semicaptive wolves. Table 1 shows details on the 21 wolves that were sampled for the presence of *L. infantum* kinetoplast DNA in tissues, including buccal mucosa, right and left ears and hair. Of the 21 samples, 14 (9 females and 5 males) belonged to specimens residing in semicaptive conditions at the Iberian Wolf Center during the period from 2018 to 2022. Until the beginning of 2022, samples were taken from the mouth, both ears and eventually from hair. From this date onwards, and once the method described in Figure 2 was implemented, samples were taken only from hair. Samples of oral mucosa and both ears were taken from the seven wild animals captured and subsequently released during routine wildlife control plans during the period 2017–2019. Five of these wolves (three males and two females) were positive for CanL, representing more than 70% positivity. With the exception of the specimen captured in Lobeznos (a 7.5-year-old male that tested positive for CanL in the ear), all animals were positive in the oral sample, which proved to be the most reliable place for sampling until its replacement in early 2022 by hair samples. Of the 14 samples taken during this period from semi-captive wolves in the Iberian Wolf Center, seven (50%) were positive in some of the samples. Of these, five of the nine females (55%) and two of the five males (40%) were PCR-positive. Overall, 12 wolf specimens (seven females and five males) of the twenty-one animals sampled alive and living in semi-captive conditions (57%) were positive for leishmaniasis.

One of the specimens sampled in the Iberian Wolf Center (identified as Brasa, a 6-year-old female) showed positivity in all samples obtained (buccal mucosa, ear and hair) and presented external symptoms of leishmaniasis (weight loss, onychogryphosis, slight lymphadenopathy and alopecia around the eyes). However, a second animal (identified as Dakota, a 9-year-old female) was positive in all samples except hair and showed no evident signs of leishmaniasis. The rest of the animals sampled were hair-positive and were equally asymptomatic.

### 3.3. Occurrence of Leishmaniasis in Dead Wolves

Table 2 shows the data collected from the sampling performed on the carcasses of 18 wolves (16 males and 2 females) sampled for the presence of *L. infantum*, including oral mucosa, right and left ears and hair. Of the 18 carcasses, 14 (77.7%) were struck by motor vehicles and found as roadkill in different areas of the province of Zamora, 3 specimens (16.7%) were killed during legal hunting and 1 wolf (5.5%) was found dead as a result of wounds received in a fight (a 5-year-old male). All animals’ oral mucosa were sampled (with the exception of one specimen sampled in Palacios de Sanabria, from which only both ears were sampled); of those, six (five males and one female) were positive (33.3%) and twelve (eleven males and one female) were negative (66.6%). Two specimens were positive in hair but not in oral samples. In no cases did the condition of the animals’ remains allow us to evaluate their symptoms of leishmaniasis.

In summary, taking into account the wolves that were positive for *Leishmania* in carcasses (6 of 18) and those that were positive in animals captured alive (12 of 21), the incidence of positivity was 18 of the 39 animals sampled (46.1%).

## 4. Discussion

Numerous publications mention the presence of *Leishmania* spp. in wildlife with different degrees of prevalence, depending on the geographical environment and climatic conditions suitable for the transmission vector during some season of the year (for a recent review, see [35]). Global warming is leading to the emergence of phlebotomine vectors in regions of the planet with cold continental climates. In the Iberian Peninsula, rising temperatures are causing the spread of sand flies to higher latitudes and altitudes than they had typically been found [4]. Phlebotominae are detected over longer annual periods due to earlier emergence from late February or early March and increasingly mild autumns, thereby delaying their disappearance until early December [36].

An example that confirms the importance of wildlife in the biological cycle of *Leishmania* was the outbreak of leishmaniasis caused by *L. infantum* that appeared in Madrid (Spain) in 2009–2012 due to the construction of a park near a residential area infested with hares and rabbits that caused more than 800 cases of leishmaniasis in humans [37].

The first reported case of CanL in wolves was described by Mohebali and coworkers in a seroprevalence study carried out in Iran to assess the impact of the disease in dogs [38]. In Europe, the first description of a dead grey wolf (*Canis lupus*) with visceral leishmaniasis caused by *L. infantum* was reported in central and southern parts of Dalmatia (Croatia), where the disease is endemic. The carcass of the animal, found dead in 2003, showed typical manifestations of the disease such as chronic dermatitis, generalized hair loss, scaling, skin erosions and ulcerations, cachexia, orchitis, lymphadenitis and hepatomegaly and splenomegaly, signs widely described in CanL. Parasitological diagnosis revealed the presence of found amastigotes in circulating macrophages. The PCR amplification of DNA samples was compatible with *L. infantum* [25,26]. 

The area sampled by us in the present study is one of the richest in wolves in the Iberian Peninsula due to the mountainous region of “Sierra de la Culebra”, which is an environmental heritage site and contains the largest wolf population. The Iberian Wolf Center “Félix Rodríguez de la Fuente” occupies a privileged position near this place, where it has maintained for years a stable population of 14 wolves in an area of 20 hectares where the animals are fed and watched over by specialized veterinarians and caretakers. Thanks to these facilities, we have been able to monitor their population for CanL prevalence using noninvasive sampling methods and detection based on the amplification of a specific region of *L. infantum* kinetoplast DNA via PCR. Wolf hair is a nontraumatic, easily accessible and safe site for wildlife sampling. *L. infantum* amplification in wolf hair samples was determined via PCR on an increasing number of hairs (5, 10 and 20) from a positive animal; amplification was observed from 10 hairs, and thus, the hair samples examined contained 10 individual hairs. Other non-invasive samples, such as those from buccal mucosa and ears, were in both cases collected by scraping with a swab, and they were also analyzed in this study for subsequent molecular analysis. 

According to our findings, the prevalence of CanL obtained in the Iberian Wolf Center in the period 2018–2022 was high, estimated at 50%. This percentage can be as high as 57% when we add the positives found in other animals captured alive, sampled and subsequently released (Table 1). These values are much higher than those obtained with captive wolves under the European Endangered Species Breeding Programme (EEP), which covers different localities in the Iberian Peninsula and southern France. In this study, it was confirmed that 3 wolves out of a total of 33 individuals tested (9%) were PCR-positive with low levels of *L. infantum* DNA in peripheral blood [39]. Other studies have been carried out in captive animals from zoological parks far from their natural environment [40]. In a survey carried out with wild animals, including three wolves from a zoo of southeastern Spain, one of the three wolves (33.3%) tested positive for *L. infantum* via PCR analysis performed on a skin sample [29]. One (50%) of two wolves from a zoo in Murcia (southeastern Spain) sampled in the period 2008–2017 tested positive in the spleen, liver or skin via PCR targeting the ITS2 of *L. infantum*. An analysis of kDNA sequences revealed genotype 2 derived from a single nucleotide polymorphism (SNP) in a wolf, which is the same as that found in domestic dogs [25].

In our study, *Leishmania*-positive results obtained from samples of dead wolves show an incidence of 33.3% of the carcasses analyzed in the period 2017–2022 (Table 2). Several works have sampled dead wolves to find *Leishmania* DNA. 

The results of the present work show a positivity rate to Leishmania of 6/18 samples from dead wolves, which represents an incidence of 33.3% of the carcasses analyzed in the period 2017–2022 (Table 2). In a prior study, in which spleen and blood samples were collected from 39 wolves in the period 1990–2007 from different regions of Spain [26], the mean positivity values were higher than 20.5% but with a higher incidence in Castilla y León (33.3%) than in Asturias (18.1%). In another study conducted in Asturias, blood samples and carcasses of 39 wolves collected during the period 2008–2010 from legal hunting, motor vehicle collisions or poisoning were analyzed via PCR using DNA extracted from mandibular lymph nodes. PCR analysis of the samples allowed us to determine a positivity rate of 46.2%, similar to the infection rate estimated in our study (46.1%), considering both live and dead animals. The presence of cutaneous lesions in 15 wolves was the only macroscopic alteration detected, but this could be due to coinfections with other pathogens [41]. 

Another retrospective study was also conducted in the Asturias region between 2008 and 2014 as part of an established wildlife health surveillance program. Using the remains of 102 necropsied wolves, a leishmaniasis seroprevalence of 33.3% was estimated; these values are lower than those provided in our results (46.1% considering the total rate of infection), with no differences between sex and age. A generalized positivity to *Leishmania* and an apparent increase in prevalence during this period were observed. According to these results, CanL does not seem to constitute a threat to the maintenance of Asturian wolf populations; thus, the authors propose that wolves may be useful sentinel species for the surveillance of *L. infantum* in field conditions [28].

It is very striking that Asturias, located in the north of Spain and considered non-endemic to leishmaniasis and phlebotome-free, has only a slightly lower prevalence than the province of Zamora, located 250 km further south, where the presence of phlebotomes and the occurrence of CanL in dogs is high [36]. These results are in line with the first leishmaniasis prevalence study conducted in the Piedmont region (northwestern Italy), where a 25.71% prevalence (CI95% 14.16–42.07) of wolves sampled after death by motor vehicle between 2009 and 2017 were positive for *L. infantum* as assessed using conventional PCR on spleen samples [42]. However, these values are much higher than those found in Iran in 2018, where the estimated infection rate was 10.2% [43].

From these studies, it is clear that the presence of Leishmania in wildlife, and more specifically in the wolf, is a proven fact that needs to be monitored not only at the level of systematic academic studies, but also during early disease control campaigns. Because the role of the wolf as a pathogen reservoir may be increasing with climate change, and linked with the fact that that vector is increasing its presence in higher and higher latitudes, leishmaniasis may turn into an emerging infection in larger nuclei of populations that had never been challenged before by this pathogen. In addition, in this work, we demonstrate the usefulness of hair samples for the diagnosis of this infection, which can help diagnose the infection without hunting the animal, as sometimes hair remains can be found in the habitats of these animals.

## Figures and Tables

**Figure 1 microorganisms-11-01179-f001:**
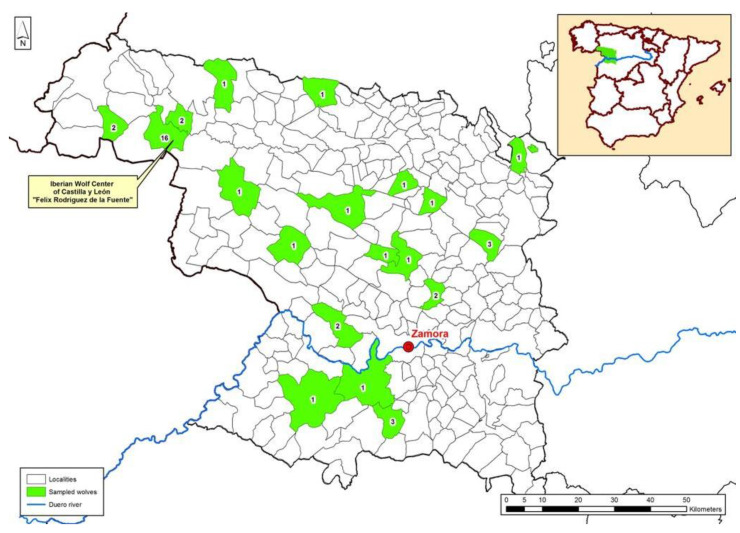
Distribution map of sampled and PCR-positive wolves (green areas) studied in Zamora (northwestern Spain) for the detection of *Leishmania* DNA from 2017 to 2022.

**Figure 2 microorganisms-11-01179-f002:**
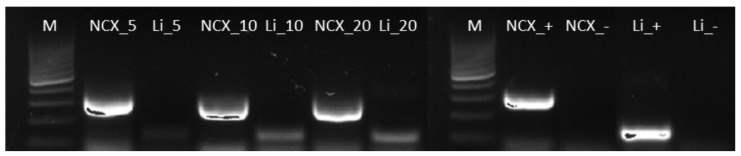
Detection of kinetoplast Leishmania DNA in hair of wolves infected with *L. infantum*. M: molecular weight ladder. PCR for the NCX1 or *L. infantum* (Li) amplification with 5, 10 or 20 hairs. In addition, positive (NCX_+ and Li_+) and negative (NCX_− and Li_−) controls were included in the reaction.

**Table 1 microorganisms-11-01179-t001:** PCR detection of *Leishmania infantum* in different samples of live captures and from the Iberian Wolf Center “Felix Rodríguez de la Fuente” (2017–2022), Zamora (Northwest Spain), including source, date sampling sex, age and site of sampling (R. ear: right ear; L. ear: left ear).

Name (Id.)	Sample Source	Date	Gender	Years	Mouth	R. Ear	L. Ear	Hair
La Albañeza	Live capture	8 November 2017	female	5	-	-	-	n.d. *
Lobeznos	Live capture	17 March 2018	male	7.5	-	+	n.d.	n.d.
Brevanas	Live capture	23 October 2018	female	0.5	+	-	-	n.d.
Cabañas	Live capture	23 October 2018	female	0.5	-	-	-	n.d.
Pereruela	Live capture	4 December 2018	female	1.5	+	-	-	n.d.
Mahide	Live capture	10 December 2018	male	0.5	+	-	-	n.d.
Brasa	Iberian Wolf Center	21 March 2018	female	6	+	+	n.d.	+
El Casal	Live capture	18 October 2019	male	0.5	+	-	-	n.d.
Dakota	Iberian Wolf Center	23 March 2021	female	9	+	+	+	-
Clarita	Iberian Wolf Center	7 May 2021	female	10	-	-	-	n.d.
Sauron	Iberian Wolf Center	7 May 2021	male	10	-	-	+	n.d.
Jara	Iberian Wolf Center	7 May 2021	female	8	-	-	-	n.d.
Oscura	Iberian Wolf Center	7 May 2021	female	10	n.d.	-	-	n.d.
Atila	Iberian Wolf Center	10 November 2021	male	9	-	-	-	n.d.
Llagu	Iberian Wolf Center	10 November 2021	male	2	-	-	-	n.d.
Robledo	Iberian Wolf Center	10 November 2021	male	6	-	-	-	n.d.
Sanabria	Iberian Wolf Center	2 February 2022	female	3	n.d.	n.d.	n.d.	+
Tera	Iberian Wolf Center	2 February 2022	female	3	n.d.	n.d.	n.d.	-
Felix	Iberian Wolf Center	3 February 2022	male	1	n.d.	n.d.	n.d.	+
Niebla	Iberian Wolf Center	3 February 2022	female	1	n.d.	n.d.	n.d.	+
Luna	Iberian Wolf Center	3 February 2022	female	1	n.d.	n.d.	n.d.	+

* n.d.: not determined.

**Table 2 microorganisms-11-01179-t002:** PCR detection of *Leishmania infantum* in different samples obtained from dead Iberian wolves (2017–2022) from the province of Zamora (Northwest Spain), including source, date sampling sex, age and site of sampling (R. ear: right ear; L. ear: left ear).

Sampling Place	Date	Cause of Death	Gender	Years	Mouth	R. Ear	L. Ear	Hair
Puebla de Sanabria 2	10 November 17	Motor vehicle	male	5	-	-	-	n.d. *
Palacios de Sanabria	10 January 18	Motor vehicle	male	3	n.d.	+	+	n.d.
Puebla de Sanabria 3	1 March 18	Motor vehicle	male	2.5	-	+	+	n.d.
Puebla de Sanabria	2 March 18	Motor vehicle	male	2.5	-	n.d.	n.d.	n.d.
Cañizo	1 October 18	Motor vehicle	female	7.5	+	-	-	n.d.
Vega del Castillo	1 February 19	Fighting	male	5	+	n.d.	n.d.	n.d.
San Cebrián	16 April 20	Motor vehicle	female	4	-	n.d.	n.d.	-
Ayoo	18 December 20	Legal hunting	male	4	-	n.d.	n.d.	+
Perilla de Castro	30 January 21	Legal hunting	male	5	-	-	-	n.d.
Cerezal	23 February 21	Motor vehicle	male	4	-	-	-	+
Santovenia	12 March 21	Motor vehicle	male	5	-	n.d.	n.d.	n.d.
Burganes	12 March 21	Legal hunting	male	4	-	n.d.	n.d.	n.d.
Torres del Carrizal 2	6 July 21	Motor vehicle	male	4	-	n.d.	n.d.	n.d.
Torres del Carrizal	5 September 21	Motor vehicle	male	1	-	-	-	n.d.
Cerezal de Aliste	1 March 22	Motor vehicle	male	4	-	n.d.	n.d.	n.d.
Puebla de Sanabria	1 March 22	Motor vehicle	male	3	-	n.d.	n.d.	n.d.
Villanueva	8 March 22	Motor vehicle	male	2	-	n.d.	n.d.	n.d.
Gallegos del Río	8 March 22	Motor vehicle	male	3	-	n.d.	n.d.	n.d.

* n.d.: not determined.

## Data Availability

Data is unavailable due to privacy.

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
