# Peer review of "Occurrence of Leishmaniasis in Iberian Wolves in Northwestern Spain"

_microorganisms, 2023, doi:10.3390/microorganisms11051179_

Round 1

Reviewer 1 Report

Dear Authors

I appreciated your paper, only minor revisions suggested are in the attached file.

Best Regards

Author Response

Responses to referee 1

I appreciate very much the revision made by your referee to our MS.

All modifications recommended by your referee have been included in the new version and are marked in yellow.

Reviewer 2 Report

This study addressed the role of leishmaniasis wolf infection in wild animals occurring in Spain. Given that this zoonotic disease can be lethal to dogs and other canines, it is very relevant to document their potential impact on captive and wild populations of wolves. Main results include a high rate of infection in individuals, reaching close to 50%. The study is interesting, with solid methods and handling protocols. The results are clearly presented. My only comment is that the authors acknowledge that possibly warmer climates in cold regions can exacerbate the presence of phlebotomine vectors transmitting the pathogen, but provide limited evidence. It would be interesting if the authors could overlap presence of phlebotomine vectors with the geographic locations of where authors sampled wolves. This can provide a strong evidence of the idea that leishmaniasis can spread to wildlife (wolves in this case) given that vectors are favored by warmer climates.     

Author Response

RESPONSE TO REFEREE 2

This study addressed the role of leishmaniasis wolf infection in wild animals occurring in Spain. Given that this zoonotic disease can be lethal to dogs and other canines, it is very relevant to document their potential impact on captive and wild populations of wolves. Main results include a high rate of infection in individuals, reaching close to 50%. The study is interesting, with solid methods and handling protocols. The results are clearly presented. My only comment is that the authors acknowledge that possibly warmer climates in cold regions can exacerbate the presence of phlebotomine vectors transmitting the pathogen, but provide limited evidence. It would be interesting if the authors could overlap presence of phlebotomine vectors with the geographic locations of where authors sampled wolves. This can provide a strong evidence of the idea that leishmaniasis can spread to wildlife (wolves in this case) given that vectors are favored by warmer climates.  

Thank you for your interest in our work. In fact, the work was conceived for the determination of Leishmania DNA in live wolves from the Iberian Wolf Center and completed with the presence of the parasite in carcasses of dead animals collected during the same period of time in a larger surrounding region. We have relied on the evidence of Phlebotomines as well as on bibliographic level and it is shown that these insects (both Phlebotomus perniciosos and Phlebotomus ariasi are present in the region) [1] (we have substituted reference 4 in old MS by this more recent reference). However, as it is not the only MS that we are willing to publish, a limited campaign of capture of insects in the Iberian Wolf Reserve is already underway.

  1. Bravo-Barriga D, Ruiz-Arrondo I, Peña RE, Lucientes J, Delacour-Estrella S. Phlebotomine sand flies (Diptera, Psychodidae) from Spain: an updated checklist and extended distributions. Zookeys. 2022 Jun 17;1106:81-99.

Reviewer 3 Report

Please consider to modify spelling of “sand flies” or “sand fly” throughout whole text it should always be written with two separate words.

Line 23-28: “This article describes for the first time the presence of Leishmania-positive wolves in the vicinity of the Sierra de la Culebra (province of Zamora, northwestern Spain), a protected sanctuary for this canid species, and discusses their possible role as a reservoir of the disease, as a secondary consequence of globalization and climate change. Taking into account the Leishmania-positive wolves found in carcasses (6 of 18) and those positive in animals captured alive (12 of 21), the positivity rate was 18 of the 39 animals sampled (46.1%)”, please change this abstract section accordingly to what mentioned for Line 57 comment below. Consider also to add results concerning set-up of PCR-based method for L. infantum DNA in wolf hair in the abstract.

Line 27-28: “the Leishmania-positive wolves found in carcasses (6 of 18) and those positive in animals captured alive (12 of 21)” Please consider to remove this sentence it is too specific for an abstract.

Line 49-50: “In Italy, a prevalence of up to 14% has been detected in the 49 Campania region”, There are no supporting references.

Line 57: “which may suffer from or be reservoirs of the disease” To be considered as reservoirs of the disease should be demonstrated at least their ability to infect the vectors with a xenodiagnosis done on infected animals, or to type the strain isolated from wild animals and demonstrate to be the same as in humans found in the same endemic area. Please change the sentence accordingly.

Line 59-61: “The incorporation of PCR as a diagnostic technique has facilitated the detection of positive animals of different species in wildlife mammals”, however, this technique sometimes overestimates positive hosts because it is able to detect even small DNA traces of a no longer viable parasite, it would have been better to have the possibility to confirm diagnoses with a different technique (serological, culture or xenodiagnoses).

Line 72-74: “Last but not least” replace with finally, “the importance of CanL should not only be addressed from the point of view of the suffering of infected animals, but as a real public health problem that may worsen with global warming”, please reconsider this sentence in light of the previous remarks.

Line 76-78: It would have been better to include other techniques to confirm the diagnosis, sometimes PCR can give positives that then do not confirm over time

Line 82: “Origin of the animals” This paragraph is very interesting, perhaps too long it is not clear how useful the information reported is for the purpose of the study please clarify.

Line 108-111: “Management and veterinary care are carried out daily, with direct observations being made every day. In addition, once a year, each animal is individually checked by veterinarians, who examine the condition of the animals and take biological samples for further studies”, It would also be interesting to report the results of Laboratory analysis carried out by veterinarians on wolves biological samples and compare them with the results obtained from the present study. At least add the medical history or eventual skin lesions presence reported. Report the results of the checks carried out if data cannot be used, please remove the sentence.

Line 135: “but also”, typo, please remove.

 Line 148-156: “All samples were analyzed by qualitative PCR after amplification of a 131-base pair 148 (bp) fragment of the kinetoplast minicircle of L. infantum using the following pair primers 149 of pair of primers (Forward: 5’ -CCCAAACTTTTCTGGTCCTC – 3’; Reverse: 5´- 150 TTACACCAACCCCCAGTTTC – 3’) corresponding to a conserved region of DNA mini- 151 circle of L. infantum kinetoplast. As internal control, to confirm that cells had been col- 152 lected with the swab, a fragment of 303 bp of the gene coding the Na+ /Ca2+ exchanger from 153 Canis lupus (NCX1) was also amplified in the buccal mucosa and inner ear samples using 154 the pair of primers previously described [33] (Forward 5′ - CCTAGGTCTCCTGCAG- 155 TGAAGT -3′; Reverse: 5′ - CCAAGACCCTTCCTTTGGA-3′)”, please specify whether it is a protocol newly developed for this study or provide references.

Line 171-173: “a quantity-dependent 131-bp band was amplified from L. 171 infantum DNA when the pair of primers corresponding to the kinetoplast DNA were 172 used”, the different quantity in the bads reporting 5, 10, 20 wolves hair is not very clear.

Line 178: “L. infantum”, please consider to change in italics throughout the text species name.

Line 189: “Fig. 2”, any Figure 2 is reported in the text.

Line 190: “On the one hand”, please remove.

Line 228-230: “Numerous publications mention the existence of CanL in wildlife with different degrees of prevalence, depending on the geographical environment that includes climatic conditions suitable for the development of the transmission vector during some season of the year – for a recent review see [34]”, please reconsider this sentence meaning is not clear.

Line 234: “Phlebotomes”, please replace with Phlebotominae.

Line 235: “cyclical periods”, please consider to replace with a more clear definition.

Line 237: “corpses of” please consider to remove.

Line 241-243: “In addition to the detrimental role that CanL can play in wildlife, the question has been raised as to whether these animals may act as a reservoir for the disease that can eventually be transmitted to humans and become a serious public health problem.”, please consider to make the sentence clearer and mostly. This sentence raises a gap in the study, to demonstrate the role of wild wolves as reservoir in the studies area, it would have been necessary to carry out entomological catches to isolate and type Leishmania strain in sand flies, as well as in vertebrates and highlight their correspondence.

Line 256-257: “PCR amplification of DNA samples was compatible with L. infantum (MON-1 strain)”, which PCR protocol allows to recognize Leishmania strain? Please specify.

Line 266-269: “According to our findings, 266 the prevalence of CanL obtained in the Iberian Wolf Center in the period 2018-2022 was high, estimated at 50%. This percentage can be as high as 57% when we add the positives found in other animals captured alive, sampled and subsequently released (Table 1)”, clarify percentage cited in the text which does not correspond to the data reported in Table 1.

Line 316-317: “is a proven fact that needs to be monitored not only at the level of systematic academic studies, but also by environmental services”, please clarify this sentence. What do you mean with environmental services?

In results section the suggestion is to merge the two paragraphs “Occurrence of leishmaniasis in living and death wolves” or clarify the reason for splitting them.

I strongly suggest shortening discussion especially the part related to retrospective studies, and emphasize results of set-up PCR-based method for L. infantum DNA in wolf hair.

Author Response

REPONSES TO REFEREE 3

The responses to this referee are marked in green color in the revised MS

Please consider to modify spelling of “sand flies” or “sand fly” throughout whole text it should always be written with two separate words.

The spelling of sand fly or sand flies was changed throughout the new version

ABSTRACT

Line 23-28: “This article describes for the first time the presence of Leishmania-positive wolves in the vicinity of the Sierra de la Culebra (province of Zamora, northwestern Spain), a protected sanctuary for this canid species, and discusses their possible role as a reservoir of the disease, as a secondary consequence of globalization and climate change. Taking into account the Leishmania-positive wolves found in carcasses (6 of 18) and those positive in animals captured alive (12 of 21), the positivity rate was 18 of the 39 animals sampled (46.1%)”, please change this abstract section accordingly to what mentioned for Line 57 comment below. Consider also to add results concerning set-up of PCR-based method for L. infantum DNA in wolf hair in the abstract.

This sentence has been changed to: “This work describes for the first time the presence of wolves positive for Leishmania in the vicinity of the Sierra de la Culebra (Zamora province, northwestern Spain), a protected sanctuary of this canid species, by PCR amplification of L. infantum DNA from different non-invasive samples such as buccal mucosa, both ears and hair. In addition to live animals (21), samples from carcasses of mainly roadkill animals (18) were also included and analyzed using the same technique, obtaining a positivity rate of 18 of the 39 wolves sampled (46.1%) regardless of their origin. Lines 23 to 29 in the new version.

Line 27-28: “the Leishmania-positive wolves found in carcasses (6 of 18) and those positive in animals captured alive (12 of 21)” Please consider to remove this sentence it is too specific for an abstract.

In the new version this sentence was removed.

INTRODUCTION

Line 49-50: “In Italy, a prevalence of up to 14% has been detected in the 49 Campania region”, There are no supporting references.

Thanks for this comment. We have included reference [9] in line 50 of the new MS to support this sentence in new MS. Note that this change implies the re-numbering of the MS.

Line 57: “which may suffer from or be reservoirs of the disease” To be considered as reservoirs of the disease should be demonstrated at least their ability to infect the vectors with a xenodiagnosis done on infected animals, or to type the strain isolated from wild animals and demonstrate to be the same as in humans found in the same endemic area. Please change the sentence accordingly.

This is an important question to be raised. There are some descriptions of CanL in wild fauna (including wolves) that suggest its role as reservoirs of the disease. As we have not done xenodiagnosis analysis, we prefer to remove the whole sentence in the new MS to avoid misunderstandings.

Line 59-61: “The incorporation of PCR as a diagnostic technique has facilitated the detection of positive animals of different species in wildlife mammals”, however, this technique sometimes overestimates positive hosts because it is able to detect even small DNA traces of a no longer viable parasite, it would have been better to have the possibility to confirm diagnoses with a different technique (serological, culture or xenodiagnoses).

We have changed the sentence from the old MS by this other according to referee´s suggestions (lines 58-61 new MS): “The incorporation of PCR as a diagnostic technique has facilitated the detection of wildlife animals that have been in contact with the parasite in some way, which in many cases has been confirmed by the use of complementary immunological and parasitological techniques”

Line 72-74: “Last but not least” replace with finally, “the importance of CanL should not only be addressed from the point of view of the suffering of infected animals, but as a real public health problem that may worsen with global warming”, please reconsider this sentence in light of the previous remarks.

According to the referee’s suggestions (Lines 71-73 new MS), “Last but not least” is replaced by “Finally,” however, we consider the importance of the rest of the sentence as CanL is always a public health problem regardless of the animal, wild or domestic, that suffers from it.

Line 76-78: It would have been better to include other techniques to confirm the diagnosis, sometimes PCR can give positives that then do not confirm over time

We agree with the reviewer. The serodiagnosis by indirect ELISA was included initially as a complementary technique to confirm the presence of the parasite but also the infection. Unfortunately, due to limitation with the final funding it was not possible to include this methodology in the current study.

MATERIALS AND METHODS

Line 82: “Origin of the animals” This paragraph is very interesting, perhaps too long it is not clear how useful the information reported is for the purpose of the study please clarify.

According to the referee’s sugestions, the paragraph has been shortened accordingly (Lines 81-102 new MS).

Line 108-111: “Management and veterinary care are carried out daily, with direct observations being made every day. In addition, once a year, each animal is individually checked by veterinarians, who examine the condition of the animals and take biological samples for further studies”, It would also be interesting to report the results of Laboratory analysis carried out by veterinarians on wolves biological samples and compare them with the results obtained from the present study. At least add the medical history or eventual skin lesions presence reported. Report the results of the checks carried out if data cannot be used, please remove the sentence.

Unfortunately we do not have permission to include in this study the results of analyses performed for different purposes other than the detection of Leishmania. This is why we have removed this sentence from the manuscript.

However, lines 193 to 199 (new MS) include a brief description of the symptoms found in two CanL-compatible wolves: “One of the specimens sampled in the Iberian Wolf Center (identified as Brasa, a 6-year-old female) showed positivity in all the samples obtained (buccal mucosa, ear and hair) and presented external symptoms of leishmaniasis (weight loss, onychogryphosis, slight lymphadenopathy and alopecia around the eyes). However, a second animal (identified as Dakota, a 9-year-old female) was positive in all samples except hair and showed no evident signs of leishmaniasis. The rest of the animals sampled were hair positive and were equally asymptomatic”

Line 135: “but also”, typo, please remove.

The typo has been removed from new MS

 Line 148-156: “All samples were analyzed by qualitative PCR after amplification of a 131-base pair 148 (bp) fragment of the kinetoplast minicircle of L. infantum using the following pair primers 149 of pair of primers (Forward: 5’ -CCCAAACTTTTCTGGTCCTC – 3’; Reverse: 5´- 150 TTACACCAACCCCCAGTTTC – 3’) corresponding to a conserved region of DNA mini- 151 circle of L. infantum kinetoplast. As internal control, to confirm that cells had been col- 152 lected with the swab, a fragment of 303 bp of the gene coding the Na+ /Ca2+ exchanger from 153 Canis lupus (NCX1) was also amplified in the buccal mucosa and inner ear samples using 154 the pair of primers previously described [33] (Forward 5′ - CCTAGGTCTCCTGCAG- 155 TGAAGT -3′; Reverse: 5′ - CCAAGACCCTTCCTTTGGA-3′)”, please specify whether it is a protocol newly developed for this study or provide references.

The first primer pair is described here for the first time, but the pair used to amplify the NCX1 gene was described previously by other authors, as indicated in the text by the reference 33.

Lines 139-141 new MS are rewritten to: “All samples were analyzed by qualitative PCR after amplification of a 131-base pair (bp) fragment of the kinetoplast minicircle of L. infantum using the following newly synthesized primer pairs”

RESULTS

Line 171-173: “a quantity-dependent 131-bp band was amplified from L. 171 infantum DNA when the pair of primers corresponding to the kinetoplast DNA were 172 used”, the different quantity in the bads reporting 5, 10, 20 wolves hair is not very clear.

We have improved the image quality of Figure 2 in order to to facilitate the visualization of the bands in the new uploaded version.

Lines 164-165 new MS: The sentence has been changed according to referee’s suggestions: “This figure shows that the amplified band was observed from 10 hairs onwards. Therefore, with the rest of the samples it was decided to analyze a total of 10 hairs per animal.”

Line 178: “L. infantum”, please consider to change in italics throughout the text species name.

These typos have been amended in new MS

Line 189: “Fig. 2”, any Figure 2 is reported in the text.

The typo has been removed from new MS

Line 190: “On the one hand”, please remove.

The typo has been removed from new MS

DISCUSSION

Line 228-230: “Numerous publications mention the existence of CanL in wildlife with different degrees of prevalence, depending on the geographical environment that includes climatic conditions suitable for the development of the transmission vector during some season of the year – for a recent review see [34]”, please reconsider this sentence meaning is not clear.

The text has been clarified according to referee´s suggestions: Lines 219-222 new MS: Numerous publications mention the presence of Leishmania spp. in wildlife with different degrees of prevalence, depending on the geographical environment and climatic conditions suitable for the transmission vector during some season of the year – for a recent review see [35]”

Line 234: “Phlebotomes”, please replace with Phlebotominae.

The typo has been removed from new MS (Line 227 new MS)

Line 235: “cyclical periods”, please consider to replace with a more clear definition.

The sentence has been modified to “Phlebotominae are detected in longer annual periods” (Line 227 new MS)

Line 237: “corpses of” please consider to remove.

The typo has been removed from new MS (Line 230 new MS)

Line 241-243: “In addition to the detrimental role that CanL can play in wildlife, the question has been raised as to whether these animals may act as a reservoir for the disease that can eventually be transmitted to humans and become a serious public health problem.”, please consider to make the sentence clearer and mostly. This sentence raises a gap in the study, to demonstrate the role of wild wolves as reservoir in the studies area, it would have been necessary to carry out entomological catches to isolate and type Leishmania strain in sand flies, as well as in vertebrates and highlight their correspondence.

Although the objective of this work has not been to demonstrate the role of the wolf as a reservoir of the disease, but the occurrence of Leishmania in wolves, we consider that it is important to hypothesize that this could be the cause, under optimal circumstances, for the transmission of the parasite through different hosts. However, and due to the suggestions of the referee, we decided to delete the whole sentence

Line 256-257: “PCR amplification of DNA samples was compatible with L. infantum (MON-1 strain)”, which PCR protocol allows to recognize Leishmania strain? Please specify.

We agree with the referee’s suggestions and we have removed MON-1 strain since it does not seem that the PCR described by these authors is strain specific. (Line 245 new MS)

Line 266-269: “According to our findings, the prevalence of CanL obtained in the Iberian Wolf Center in the period 2018-2022 was high, estimated at 50%. This percentage can be as high as 57% when we add the positives found in other animals captured alive, sampled and subsequently released (Table 1)”, clarify percentage cited in the text which does not correspond to the data reported in Table 1.

Overall, 12 wolf specimens (7 females and 5 males) of the 21 animals sampled alive and living in semi-captive conditions (57%) were positive for leishmaniasis.

In the lines 189 to 194 of the new MS (RESULTS) we have clarified “in the Iberian Wolf Center” (Line 190): On the other hand, of the 14 samples taken during this period from semi-captive wolves in the Iberian Wolf Center, 7 (50%) were positive in some of the samples. Of these, 5 of the 9 females (55%) and 2 of the 5 males (40%) were PCR positive. Overall, 12 wolf specimens (7 females and 5 males) of the 21 animals sampled alive and living in semi-captive conditions (57%) were positive for leishmaniasis.

Line 316-317: “is a proven fact that needs to be monitored not only at the level of systematic academic studies, but also by environmental services”, please clarify this sentence. What do you mean with environmental services?

According to referee 3, we include this sentence in line 314 new MS “…during early disease prevention campaigns.”

In results section the suggestion is to merge the two paragraphs “Occurrence of leishmaniasis in living and death wolves” or clarify the reason for splitting them.

In our opinion, it is easier to read the MS by dividing this section into live and dead animals.

I strongly suggest shortening discussion especially the part related to retrospective studies, and emphasize results of set-up PCR-based method for L. infantum DNA in wolf hair.

We appreciate very much this suggestion, and we have included the following sentence in lines 256-262  new MS: “Wolf hair is a non-traumatic, easily accessible and safe site for wildlife sampling. L. infantum amplification in wolf hair samples was determined by PCR on an increasing number of hairs (5, 10 and 20) from a positive animal; amplification was observed from 10 hairs, thus the hair samples examined contained 10 individual hairs. Other non-invasive samples, such as those from buccal mucosa and ears, in both cases collected by scraping with a swab, were also analyzed in this study for subsequent molecular analysis”.